# Historical Changes and Future Trajectories of Deforestation in the Ituri-Epulu-Aru Landscape (Democratic Republic of the Congo)

**Joël Masimo Kabuanga** [1,2,*] **, Onésime Mubenga Kankonda** [1,3]**, Mehdi Saqalli** [4]**, Nicolas Maestripieri** [5]**, Thomas Mumuni Bilintoh** [6]**, Jean-Pierre Mate Mweru** [7]**, Aimé Balimbaki Liama** [8]**, Radar Nishuli** [8] **and Landing Mané** [9]

1   Département d'Aménagement des Écosystèmes, Faculté de Gestion des Ressources Naturelles Renouvelables, Université de Kisangani, Kisangani P.O. Box 2012, Democratic Republic of the Congo; onesime.kankonda@unikis.ac.cd
2   Royal Zoological Society of Antwerp (KMDA) & JURISTRALE, 25, avenue de l'OUA-Ngaliema, P.O. Box 16576 Kin 1, Kinshasa, Democratic Republic of Congo
3   Centre de Surveillance de la Biodiversité, Université de Kisangani, Kisangani P.O. Box 2012, Democratic Republic of the Congo
4   CNRS (UMR 5602 GEODE) Maison de la Recherche de l'Université du Mirail, 5, Allées A. Machado, CEDEX 1, 31058 Toulouse, France; mehdi.saqalli@univ-tlse2.fr
5   TerraNIS, 12 Avenue de l'Europe, 31520 Ramonville, France; nicolas.maestripieri@terranis.fr
6   School of Geography, Clark University, Worcester, MA 01610, USA; tbilintoh@clarku.edu
7   Ecole Post-Régionale d'Aménagement et de Gestion Intégrés des Forêts et des Territoires Tropicaux (ERAIFT), Université de Kinshasa, Kinshasa P.O. Box 15.373, Democratic Republic of the Congo; jp.mate@eraift-rdc.org
8   Institut Congolais pour la Conservation de la Nature (ICCN), 13 Avenue des Cliniques, Kinshasa P.O. Box 868 Kin1, Democratic Republic of the Congo; aime.joseph@yahoo.fr (A.B.L.); radarnishuli3@gmail.com (R.N.)
9   Satellite Observatory of the Forests of Central Africa, 14, Sergent Moke—Q/Socimat, Concession Safricas, Kinshasa, Democratic Republic of the Congo; lmane@osfac.net
*   Correspondence: mkabuang@hotmail.com; Tel.: +243-826-369-021

**Abstract:** The Ituri-Epulu-Aru landscape (IEAL) is experiencing deforestation and forest degradation. This deforestation is at the root of many environmental disturbances in a region characterized by endemism in biodiversity. The importance of this article is to provide useful information for those who wish to discuss a model that can be replicated for other territories affected by deforestation and changes in natural and anthropogenic forest structure. This article focuses on the triangulation of spatialized prospective scenarios in order to identify future trajectories based on the knowledge of historical dynamics through the diachronic analysis of three satellite images (2003–2010–2014–2016). The scenarios were designed in a supervised model implemented in the DINAMICA EGO platform. The three scenarios: business as-usual (BAU), rapid economic growth (REG) and sustainable management of the environment (SME), extrapolating current trends, show that by 2061 this landscape will always be dominated forests (+84%). Old-growth forests occupy 74.2% of the landscape area in the BAU scenario, 81.4% in the SEM scenario and 61.2% in the REG scenario. The SEM scenario gives hope that restoration and preservation of biodiversity priority habitats is still possible if policy makers become aware of it.

**Keywords:** land use change; modeling; scenario; deforestation; DINAMICA EGO; PFBC landscapes; Democratic Republic of the Congo

## 1. Introduction

Deforestation is one of the main environmental problems in the Democratic Republic of the Congo (DRC) [1]. Studies show that deforestation and forest degradation cause disturbances at several levels, such as biodiversity loss, soil erosion and global warming [2].

Indeed, these two processes lead to the modification of the composition and configuration of forest landscapes [3]. Old-growth forest is considered as the priority habitat for biodiversity because it corresponds with the undisturbed natural ecosystem [4,5]. Its replacement by other land uses is therefore of significant ecological concern [6]. Moreover, deforestation and habitat loss represent complex phenomena linked to several causes, in particular the expansion of agriculture, the extension of infrastructure, logging, economic, demographic, cultural, technological, political factors and institutional establishment [7–9]. However, the influence of these factors depends on their intensity and the duration of their pressure [10].

Although quantitative and qualitative studies on the influence of various causes remain rare, the literature agrees that shifting slash-and-burn agriculture is the main driver of deforestation in DR Congo [11,12]. Knowledge from studies of land use and occupation changes is available at the national level [12,13], but it remains less numerous at the provincial and local level, particularly in landscape conservation [13]. In the Ituri-Epulu-Aru landscape (IEAL), studies on change stop at estimating forest area and deforestation rates [2,14,15]. Moreover, studies on the spatiotemporal modeling of forests have recently been produced however very few have been developed and applied at the scale of a conservation landscape [1].

The Ituri-Epulu-Aru landscape is one of twelve conservation landscapes under the Congo Basin Forest Partnership (CBFP). This landscape is mainly dominated by tropical rainforests [15]. Furthermore, it abounds in an exceptional biodiversity including in particular more than 1192 species of plants, 62 species of large mammals (including the extremely rare okapi, the forest elephant and the chimpanzee) and 312 species of birds [15,16]. Deforestation and forest degradation are the main threats to this biodiversity.

The changes in land cover and use across the Ituri-Epulu-Aru landscape are poorly understood and poorly documented [14]. Yet, it is the sum of local dynamics that determines change at the national, regional and global scale [17]. Consequently, the expansion of deforestation raises a series of questions regarding the evolution of priority habitats for biodiversity, its impact on the composition and configuration of the landscape, the role of the dominant factors in the past dynamics and the possible future devastation of forests in the short, medium and long term.

Remote sensing is useful for monitoring vegetation [18]. However, the mapping of land use by remote sensing remains a methodological challenge in the tropical region, given the heavy cloudiness there. Access to satellite images also remains limited. In the Ituri-Epulu-Aru landscape, many institutions working in the management of natural resources rely on cartographic material from national studies [2] regional or global due to lack of technology or financial constraints [13]. However, the definition of the legend or the observation time may not always meet the expectations of managers.

The interest of this study was to simulate deforestation in the future based on present and past deforestation. In addition, the simulations were analyzed in contrasting scenarios in order to plan future actions to fight against deforestation [7,8,19–21].

## 2. Materials and Methods

### 2.1. Study Area

The Ituri-Epulu-Aru landscape (2°37′022″–0°31′030″ N, 27°34′034″–30°00′039″ E, 40,862 km$^2$) is one of the twelve CBFP landscapes (Figure 1). It is located in the northeastern part of the Democratic Republic of the Congo. Most of the landscape is located in Ituri province (in the administrative territories of Mambasa, Irumu and Djugu). A part of the landscape is included in the province of Haut-Uélé (territories of Wamba and Watsa). Another part also affects the province of North Kivu from where part of the population leaves and affects the landscape.

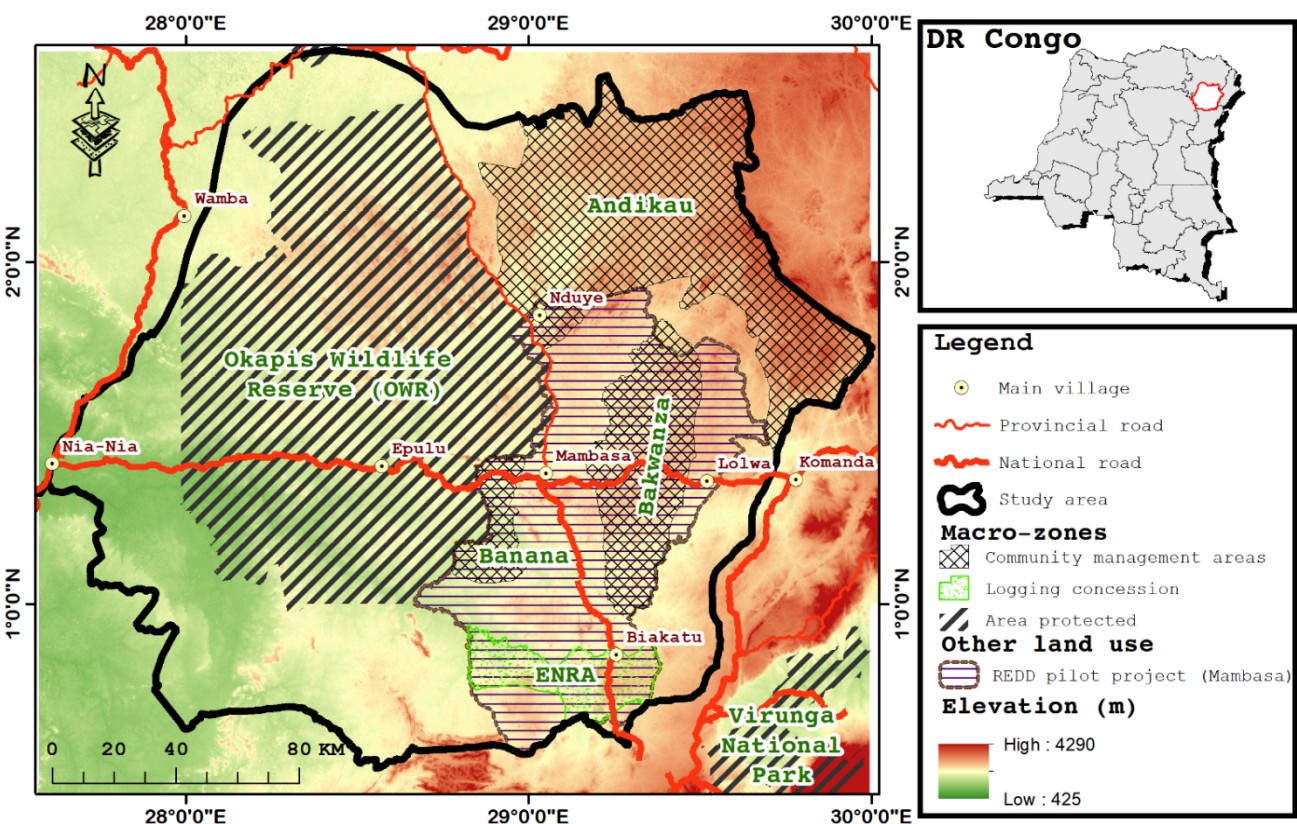

**Figure 1.** Geographical and topographical context of the study area.

EIAL is characterized by its high biodiversity and number of endemic species. The biophysical occupation of the Ituri-Epulu-Aru landscape is mainly dominated by dense semievergreen dryland to closed canopy forests. These forests include, in particular, the monodominant forests with *Gilbertiodendron dewevrei* and the mixed forests in which no species is predominant. In the extreme northeast of the landscape there is the semideciduous forest, the canopy of which is mainly composed of heliophilous species such as *Entandrophragma* spp. and *Khaya anthotheca*, *Albizia* spp. [15,22,23]. Secondary forests and the rural complex are very often along the roads. The region's economic activities are shifting slash-and-burn agriculture, artisanal and semi-industrial mining, artisanal and industrial logging and animal husbandry. The agricultural area is divided into two distinct sectors: the hut gardens and the fields far from the villages. The agriculture practiced by the groups that traditionally live in the forest is based on a rotation of two years of crops and ten years of fallow. The fields are small, generally less than 2 ha, and represent only a small proportion of the agricultural mosaic. Recent immigrants practice more intensive agriculture, with larger fields, shorter fallow periods, and greater clearing of old-growth forest [22,23].

Land use at the landscape scales (Figure 1) includes the Okapis Wildlife Reserve (OWR) (13,720 km$^2$), the Mai-Tatu Community Reserve (proposed), a logging concession Enzyme Refines Association (ENRA) (520 km$^2$) and three community management zones: Banana (575 km$^2$), Andekau (6973 km$^2$) and Bakwanza (2181 km$^2$) [22,23].

### 2.2. Data Used

2.2.1. Satellite Images

Satellite images used for land use dynamics in the Epulu-Ituri-Aru landscape are annual CARPE composites of 0.025 degree resolution. These composites come from Landsat TM, ETM + and OLI images (respectively, Thematic Mapper, Enhanced Thematic Mapper plus and Operational Land Imager). These composites are made up of four spectral bands:

NIR (0.845–0.885 μm), RED (0.63–0.68 μm), SWIR1 (1.56–1.66 μm) and SWIR2 (2.1–2.3 μm). These composites have undergone atmospheric, radiometric and geometric corrections [24].

The Central Africa Regional Program for the Environment (CARPE) composites were chosen because they have no cloud cover and allow the analysis of multi-date changes. They cover all the countries of the Congo Basin and can be downloaded free of charge from the CARPE website (https://carpe.umd.edu/ (accessed on 3 September 2021)) [24]. These images are organized in square tiles of one degree. For this article, 40 tiles were used for the four dates selected (i.e., 10 tiles per date). The CBFP landscapes were created in 2002 and development works started in 2003 in the Epulu-Ituri-Aru landscape. Therefore, the year 2003 was chosen as the reference date. In addition, 2016 was chosen in alignment with a field data collection campaign. And 2010 is the year that roughly halves the observation period (2003 and 2016). The year 2014 was chosen for the validation of the spatialized prospective model. Indeed, 2014 is relatively close to 2010 and 2016 and far enough away from 2003; an ideal time step for validation [25–27].

To ensure multi-temporal comparability, a series of preprocessing were useful. First, the rectified images were projected in the same reference coordinate system: WGS 84, UTM zones 35 North. Then, for each spectral band, a mosaic of tiles was created in the chosen years. Radiometric shifts due to differences in acquisition dates were minimized by doing histogram equalization while taking the sharper tiles as references.

### 2.2.2. Field Data

Supervised classification generally requires a certain number of training samples and verification samples [26]. Typically, traditional search uses manual visual interpretation to get points. Thus, the sampling consisted of the selection of the objects according to the spectral profiles defined using the GPS field surveys (surveys from 20 December 2016 to 15 January 2017). Then, the training areas that were chosen, on the images after 2016, for each class correspond to areas considered unchanged (built-up areas and inselberg for example) or having signatures close to the profile of 2016. In total, 950 measurement points have been taken (Table 1). This set was split into two groups of data: 665 used for the classification of land use in 2016 (i.e., 70% geographic coordinates) and 285 points used for the validation of the 2016 classification.

**Table 1.** Description of land use classes.

| Land Cover | Code | Number of Points | Description | Sources |
|---|---|---|---|---|
| Old-growth forest | Pf | 257 | Woody formation consists of a very dense cover of large trees. Old-growth forest can be semi-deciduous or evergreen, or even swampy. In all cases, the carpet of grasses is absent, and the forest has not undergone significant modification by human activities. The tree layer can reach 50 m in height. | [24,28,29] |
| Secondary forest | Sf | 302 | Woody formation corresponding to a stage of reconstitution of forest massifs which have undergone strong anthropogenic interventions, or which have evolved from wastelands. It usually has a strong dominance of moderately fast growing semi-heliophilic species. The tree layer generally reaches 35 m in height | [6,24,29–31] |
| Non-Forest | NF | 315 | Non-forest plant formation including wasteland, shrub savannah, land cultivated on an itinerant or intensive basis, as well as recent fallows. This class also includes areas occupied by buildings, dwellings and other high-density constructions as well as areas without vegetation with bare soil, rocky outcrops or even sandy beaches along rivers. This class is represented by the major roads and their right-of-way | [24,28,30–32] |
| Water | Ww | 76 | This class includes all bodies of water, including the Ituri River and Epulu | [13,24,28,30] |

*2.3. Methods*

The technical process can be divided into 2 steps:

- Land use and land cover (LULC) classification.
- Modeling of deforestation.

The overall technical process is shown in Figure 2.

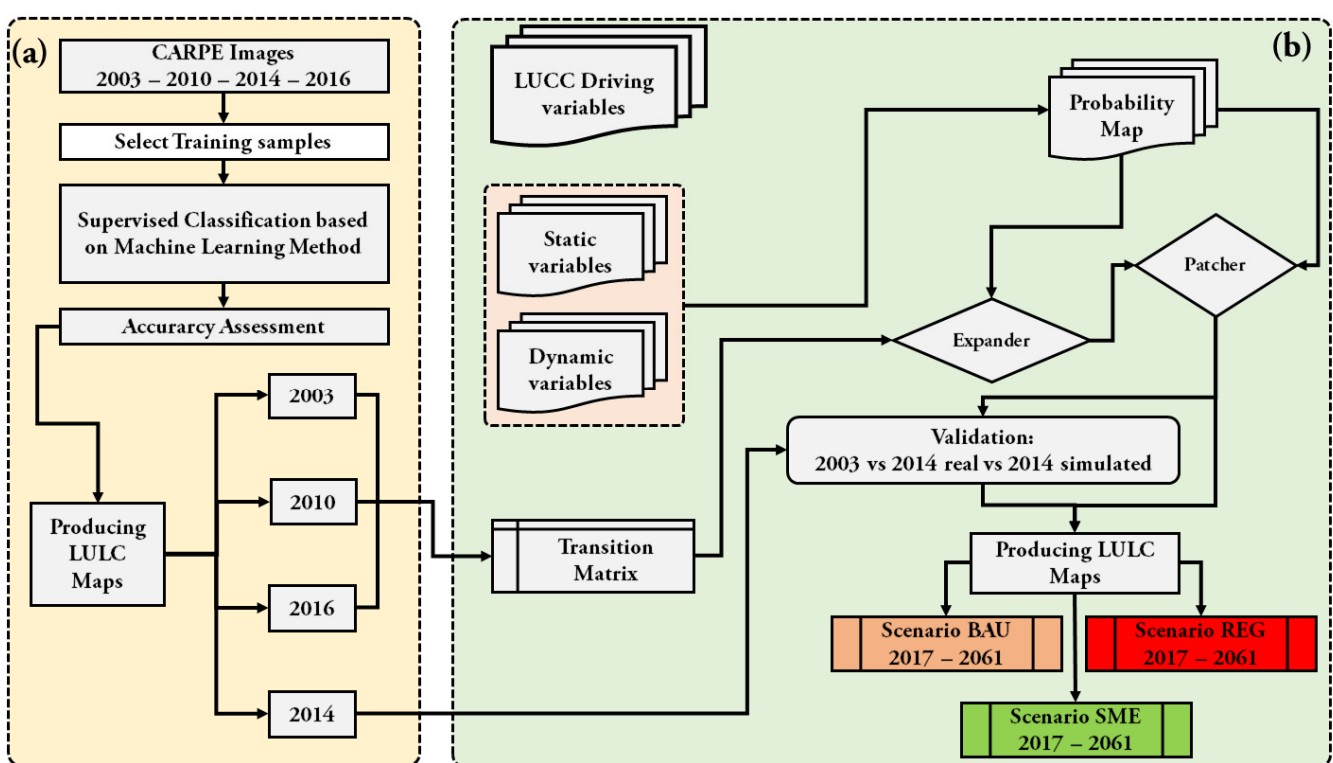

**Figure 2.** Workflow of the study: (**a**) Land use and land cover (LULC) classification; (**b**) Modeling of deforestation. Abbreviations: CARPE, The Central Africa Regional Program for the Environment; LUCC, Land use and cover changes; LULC, Land use and land cover.

2.3.1. Land Use and Land Cover (LULC) Classification

The Random Forest classification (RFC) was applied to the images of 2003, 2010, 2014 and 2016 via the R software [33] with the R package "RandomForest" [34] to obtain LULC information. RFC [35] is a supervised technique of nonparametric statistical methods [36]. RFC has been used in several studies in the past [24,32,36–43]. In the RFC, when a sample is entered into the model, each decision tree performs a separate evaluation to determine which category the sample should belong to, and the category that is most often selected is ultimately considered the category sample. The RF method can effectively reduce the uncertainty of a particular algorithm and improve the precision of the discriminant classification. The informational dimension of RF processing is larger and more complex than that of other classification algorithms.

In this study, the data entered into the RF model included the full range of raw bands of the annual composites of CARPE (RED, NIR, SWIR1 and SWIR2), Normalized difference vegetation index (NDVI), Normalized difference moisture index (NDMI), Band5/Band 4 ratio (B54R), Normalized brown ratio (NBR), SRTM products. During the study, we found that the classification accuracy of the full band combination was highest when comparing different combinations of bands. Additionally, we have found that SRTM products improve overall accuracy. The number of decision trees was set at 2000 using 70% of all samples.

### 2.3.2. Modeling of Deforestation

Spatial modeling of deforestation was made on the basis of historical changes in land use assessed between 2003 and 2016. The combination of the transition matrix (2003–2016) adapted to three scenarios: business as usual (BAU), sustainable environmental management (SEM) and rapid economic growth (REC), with maps of transition potential and explanatory factors has enabled regular prospective monitoring up to 2061 to be established using a probabilistic model designed in the DINAMICA EGO platform [10,21,44]. The deforestation simulation included: (i) selection of factors of change, (ii) transitions, (iii) exploratory analysis of deforestation factors, (iv) simulation and (v) validation.

### Selection of Variables

Variable selection eliminates overly correlated variables and contributes to the success of the modeling [9,45]. On the basis of the literature, fieldwork and general reflection made it possible to identify the variables (factors) explaining deforestation [11,14,22,23,30,46]. The variables identified were grouped into six categories (Agriculture, Economic factors, Transport, Demographic factors, Sociopolitical factors, Biophysical factors [11]. Only spatially explicit variables were retained for this study. Then, these variables were quantified in a geographic information system (Figure 3). Finally, an exploratory univariate analysis, calculating the correlation between the explanatory variables and deforestation and forest degradation, was carried out to identify the relationships between deforestation and each of the explanatory variables (Table 2).

**Table 2.** Explanatory variables of deforestation.

| Category | Variable Retained | Code | Sources |
|---|---|---|---|
| | Distance to agricultural areas | d_agri | Spatial analysis [24] |
| Agriculture | Rural complex | Comp | [24] |
| | Distance to rural complex | d_comp | Spatial analysis [24] |
| | Distances to built-up areas | d_abat | Spatial analysis [24] |
| | Distances to major center | d_gcent | Spatial analysis [24] |
| Economic factors | Forest concessions | Ccf | [47] |
| | Mining square | Mining | [47] |
| | Distance to mining squares | d_mining | Spatial analysis [47] |
| | Distance to national road | d_road1 | Spatial analysis [47] |
| Transport | Distance to provincial road | d_road2 | Spatial analysis [47] |
| | Distance to local road | d_road3 | Spatial analysis [47] |
| Demographic factors | Population density | Dens | [48] |
| | Protected areas | Ap | [49] |
| Sociopolitical factors | Agricultural zones delimited | Areaagr | [49] |
| | community management | Areamngt | [49] |
| | Elevation | Dme | [50] |
| | Slope | Slope | Spatial analysis [50] |
| Biophysical factors | Distance to watercourses | d_w | Spatial analysis [13,24,50] |
| | Distances to non-forests | d_nf | Spatial analysis [24,30,46] |
| | Distance to degraded forest | d_fd | Spatial analysis [24] |

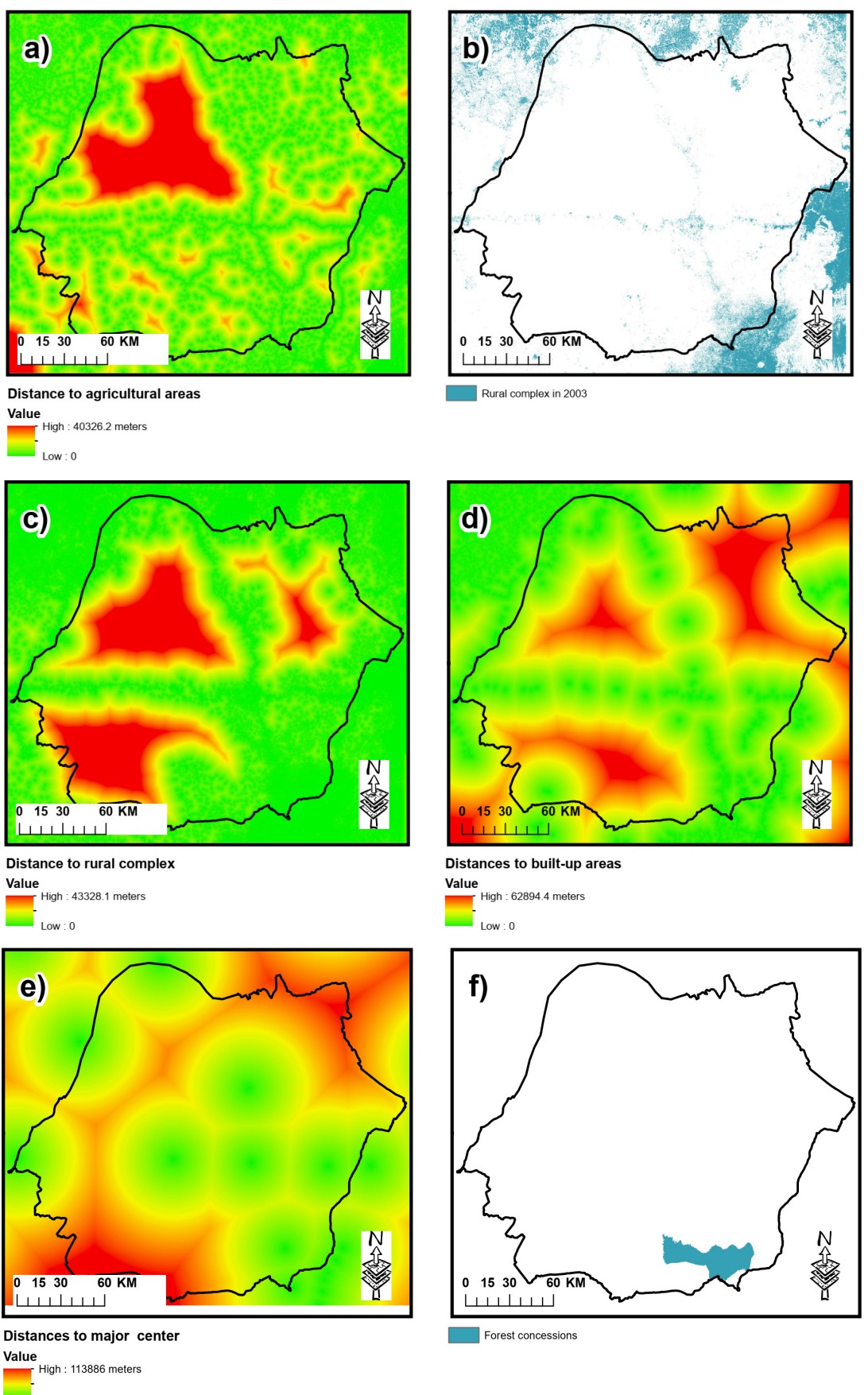

**Figure 3.** *Cont.*

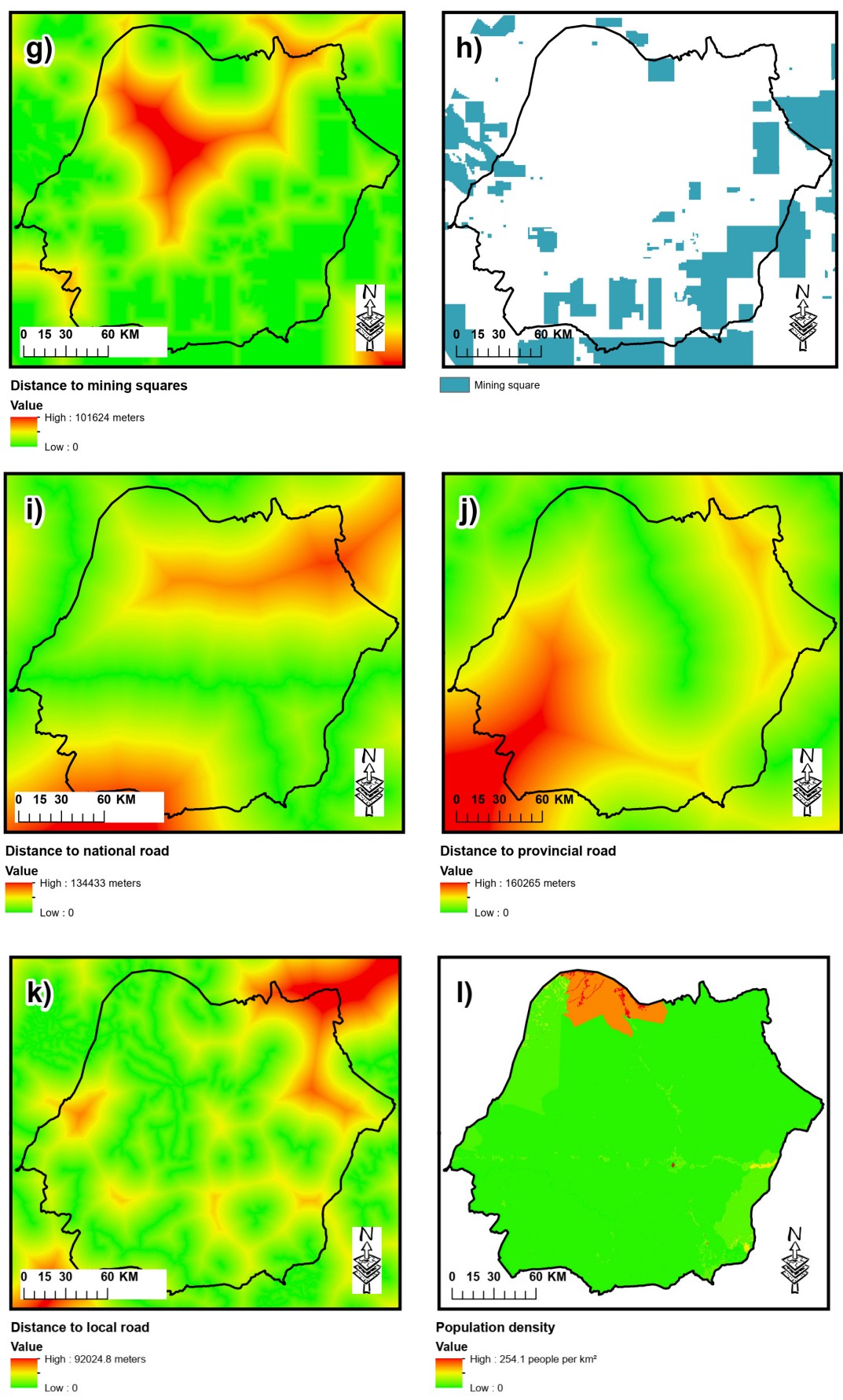

**Figure 3.** *Cont.*

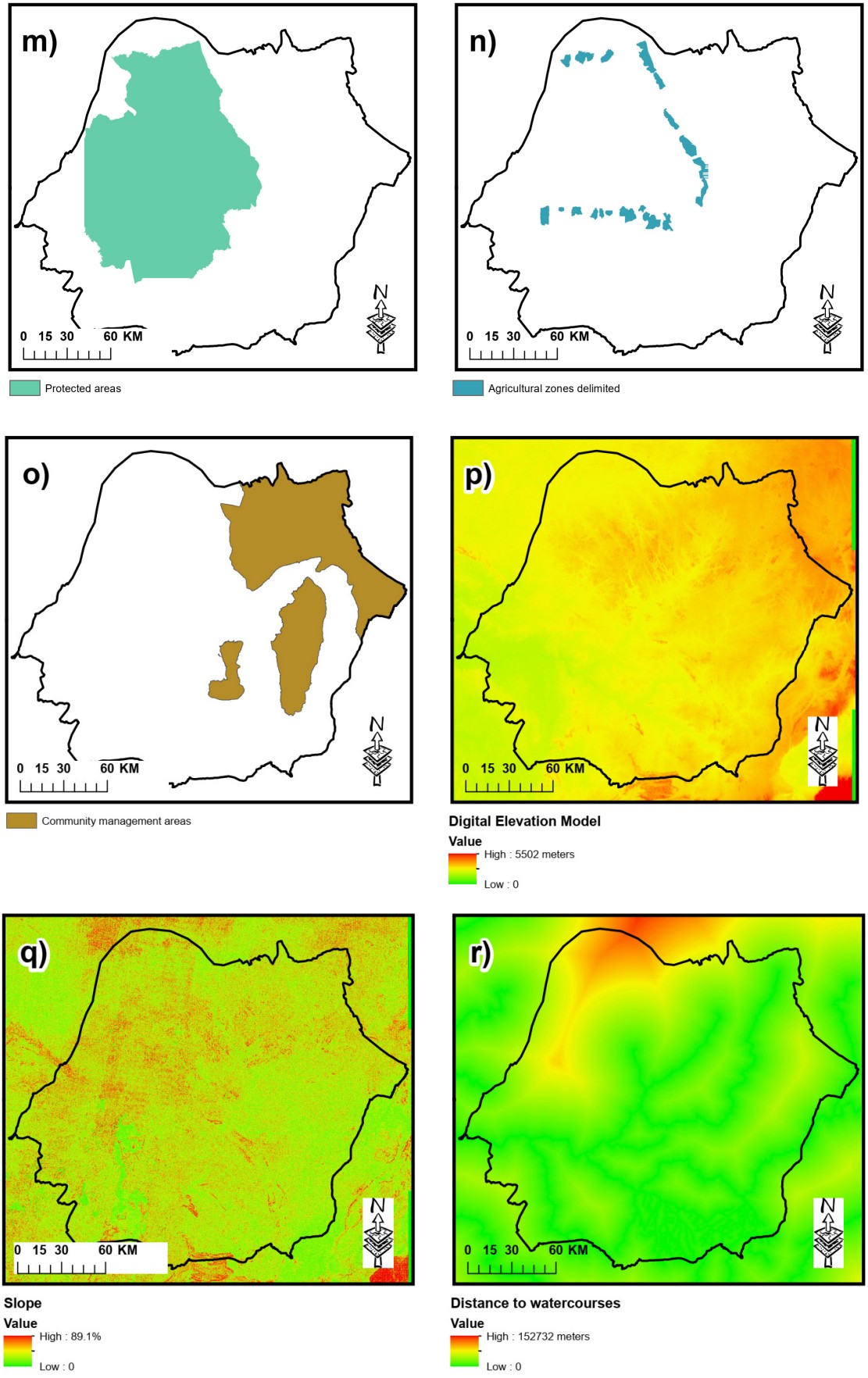

**Figure 3.** *Cont.*

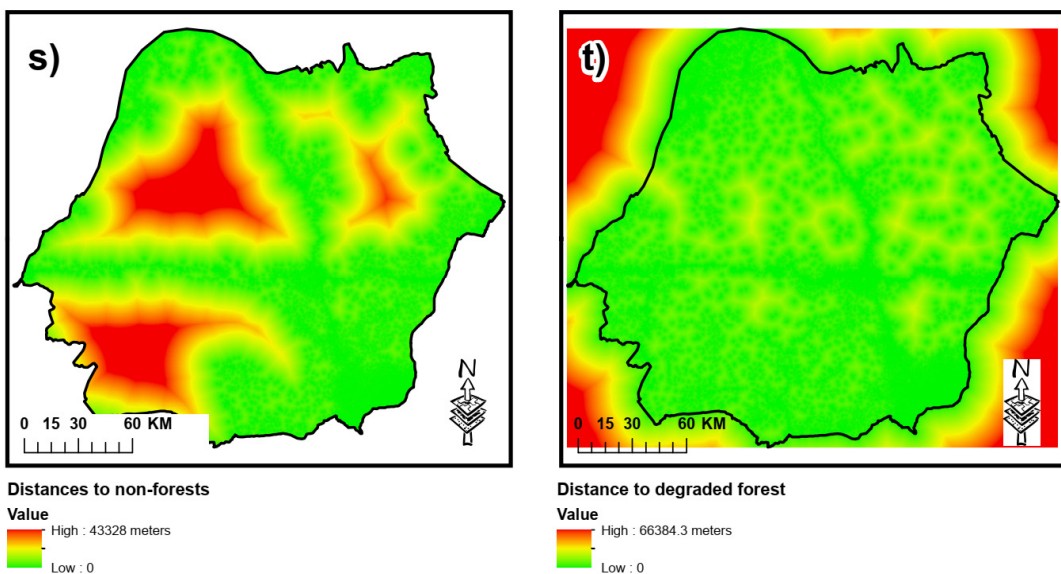

**Figure 3.** Maps of explanatory variables for deforestation. (**a**) distance to agricultural areas; (**b**) rural complex; (**c**) distance to rural complex; (**d**) distances to built-up areas; (**e**) distances to major center; (**f**) forest concessions; (**g**) distance to mining squares; (**h**) mining square; (**i**) distance to national road; (**j**) distance to provincial road; (**k**) distance to local road; (**l**) population density; (**m**) protected areas; (**n**) agricultural zones delimited; (**o**) community management; (**p**) elevation; (**q**) slope; (**r**) distance to watercourses; (**s**) distances to non-forests; (**t**) distance to degraded forest.

Transitions

The transition corresponds to the total amount of LULCC that occurred during the simulation period. The quantities of change were calculated by the Markovian method. They constitute an essential element in the simulation of changes in land cover and land use because they determine the surface area to be allocated in space according to the change probability maps and the various constraints defined [18]. Transitions modeled in this study include (a) transition from old-growth to secondary forest (degradation), (b) transition from old-growth to non-forest (deforestation), (c) transition from secondary forest to old-growth (maturation), and (d) transition from secondary forest to non-forest (deforestation).

Exploratory Analysis of the Data

When the dynamics of LULCC was modelled, weight of evidence (WoE) was applied to the transition probabilities of the project. The weight of evidence represents the influence of each variable on the spatial probability of an *i-j* transition. A work of adjustment of the weights of evidence was useful because the input data is not always entirely reliable, and some categories resulting from the operations of discretization may be nonexistent. This adjustment, requiring expert knowledge, brought relevant added value. The operation was based on the automatically calculated values and is carried out through a visualization interface made available by DINAMICA EGO. As required by the literature [18,51], no fundamental modifications were applied. The purpose of the adjustment is twofold: to model the most obvious functions (for example in the case of distances or altitude), and to adjust the values deemed unrealistic. Then, pairwise tests were performed for categorical maps to assess the independence hypothesis. The methods used are Chi2, Crammer's V index, contingency, entropy and joint uncertainty information [52]. The purpose of this step was the selection of variables because the study of the past and the present which was not the only way of explaining future deforestation. Its interest was to retain those who best contribute to the establishment of each land use class. Although there is no unanimity on the cut-off that should be used to exclude a variable, a common practice, also adopted in this study, is to choose a cut-off of 0.5 from the Crammer V index. (Measure

of the relationship between categorical variables). Above this value, the variables are correlated [21,53].

Simulation of Deforestation

The simulation of land use changes is carried out in order to facilitate decision making. The interest of this simulation lies in its ability to construct the future image of forests according to three contrasting scenarios: a "trend scenario" (business as usual, (BAU)) which starts from the hypothesis of the absence of new economic or environmental policies, a "sustainable environmental management" (SEM) scenario in which legislation and government subsidies encourage the emergence of forestry (multiplication of plantations and agroforestry) and the protection of wood resources, and finally, a "socio-economic" scenario (rapid economic growth, (REG)), i.e., acceleration of the destruction of tree and shrub plant cover and expansion of agricultural land (tendency towards disaster).

Validation

The validation of the simulation model focused on budgeting for errors and correct predictions [54,55]. In practice, this involves comparing three maps: (i) the map of the initial year (2003), (ii) the simulated map in 2014, and (iii) the one produced by satellite image classification in the same year (2014). This three map analysis shows how simulated change compares to baseline change by revealing five components [54,56,57]: (1) the reference change simulated correctly as a change (i.e., hits), (2) reference change simulated incorrectly as persistence (i.e., misses), (3) reference persistence incorrectly simulated as a change (i.e., false alarms), (4) persistence of the correctly simulated reference as persistence (i.e., correct rejections) and (5) reference change simulated incorrectly as a change to the wrong gain category (i.e., false results) (Table 3). Based on these pixels, two types of errors were evaluated in order to judge the accuracy of the overall prediction across the entire landscape. First, the quantity error (Q) was determined by the difference between false alarms and misses (Q = $|F - M|$). Finally, the allocation error (A) calculated by the difference of the total observed changes (OC = M + H) with the quantity errors [A = (F + M) − Q]. The total observed changes (OC) are given by the sum of misses and hits (OC = M + H). Also, the total predicted changes were determined by the combination of false alarms and hits (PC = F + H).

**Table 3.** Approach to error budgeting and correct predictions. 1 = Old-growth forest; 2 = Secondary forest; 3 = Non-Forest; 4 = Water.

| Comparison of Three Maps | | | | |
|---|---|---|---|---|
| **2003** | **2014** | **2014si** | **Components** | |
| 1 | 1 | 1 | Reference persistence simulated correctly as persistence | Correct rejections |
| 2 | 2 | 2 | | |
| 3 | 3 | 3 | | |
| 4 | 4 | 4 | | |
| 1 | 2 | 1 | Reference change simulated incorrectly as persistence | Misses |
| 1 | 3 | 1 | | |
| 1 | 4 | 1 | | |
| 2 | 1 | 2 | | |
| 2 | 3 | 2 | | |
| 2 | 4 | 2 | | |
| 3 | 1 | 3 | | |
| 3 | 2 | 3 | | |
| 3 | 4 | 3 | | |
| 4 | 1 | 4 | | |
| 4 | 2 | 4 | | |
| 4 | 3 | 4 | | |

**Table 3.** *Cont.*

| Comparison of Three Maps | | | | |
|---|---|---|---|---|
| **2003** | **2014** | **2014si** | **Components** | |
| 1 | 1 | 2 | | |
| 1 | 1 | 3 | | |
| 1 | 1 | 4 | | |
| 2 | 2 | 1 | | |
| 2 | 2 | 3 | | |
| 2 | 2 | 4 | Reference persistence simulated | False Alarms |
| 3 | 3 | 1 | incorrectly as change | |
| 3 | 3 | 2 | | |
| 3 | 3 | 4 | | |
| 4 | 4 | 1 | | |
| 4 | 4 | 2 | | |
| 4 | 4 | 3 | | |
| 1 | 2 | 2 | | |
| 1 | 3 | 3 | | |
| 1 | 4 | 4 | | |
| 2 | 1 | 1 | | |
| 2 | 3 | 3 | | |
| 2 | 4 | 4 | Reference change simulated | Hits |
| 3 | 1 | 1 | correctly as change | |
| 3 | 2 | 2 | | |
| 3 | 4 | 4 | | |
| 4 | 1 | 1 | | |
| 4 | 2 | 2 | | |
| 4 | 3 | 3 | | |
| 1 | 2 | 3 | | |
| 1 | 2 | 4 | | |
| 1 | 3 | 2 | | |
| 1 | 3 | 4 | | |
| 1 | 4 | 2 | | |
| 1 | 4 | 3 | | |
| 2 | 1 | 3 | | |
| 2 | 1 | 4 | | |
| 2 | 3 | 1 | | |
| 2 | 3 | 4 | | |
| 2 | 4 | 1 | | |
| 2 | 4 | 3 | Reference change simulated | |
| 3 | 1 | 2 | incorrectly as change to the wrong | Wrong Hits |
| 3 | 1 | 4 | gaining category | |
| 3 | 2 | 1 | | |
| 3 | 2 | 4 | | |
| 3 | 4 | 1 | | |
| 3 | 4 | 2 | | |
| 4 | 1 | 2 | | |
| 4 | 1 | 3 | | |
| 4 | 2 | 1 | | |
| 4 | 2 | 3 | | |
| 4 | 3 | 1 | | |
| 4 | 3 | 2 | | |

## 3. Results

### 3.1. Assessment of the Quality of Land Use Maps

The overall precision of land cover classifications in the study area from 2003 to 2016 was 0.94 ± 0.03. In detail, old-growth forest, Non-Forest and Water exhibited a higher classification accuracy than the Secondary Forest class. Their User Precision (UA) and Producer Precision (PA) values were in all cases greater than 0.75 (Table 4). The precision

was low for secondary forests, with AU of 0.8 ± 0.02 and PA of 0.78 ± 0.02. In addition, non-forests had intermediate precision values, with AU of 0.82 ± 0.03 and BP of 0.8 ± 0.02. There have been many instances in which secondary forests have been incorrectly classified as old-growth forest and non-Forest.

**Table 4.** Accuracy assessments of land cover classifications.

| Accuracy | Land Use | 2003 | 2010 | 2014 | 2016 |
|---|---|---|---|---|---|
| User | Pf | 0.91 | 0.93 | 0.89 | 0.98 |
| | Sf | 0.78 | 0.82 | 0.79 | 0.77 |
| | Nf | 0.81 | 0.82 | 0.79 | 0.85 |
| | Ww | 0.92 | 0.94 | 0.91 | 0.90 |
| Producer | Pf | 0.89 | 0.90 | 0.86 | 0.94 |
| | Sf | 0.77 | 0.79 | 0.76 | 0.81 |
| | Nf | 0.79 | 0.80 | 0.78 | 0.82 |
| | Ww | 0.90 | 0.92 | 0.89 | 0.95 |
| Over all | | 0.91 | 0.93 | 0.93 | 0.97 |

*3.2. Analysis of Historical Changes of Deforestation between 2003 and 2016*

Table 5 presents: (i) forest areas (ha), (ii) deforested areas between 2003–2010 and between 2010–2016 (ha) and (iii) observed deforestation rates by axis (%). For all dates, old-growth forest represents more than 3,600,000 ha. The most deforestation event is observed between 2010–2016. It is the deforestation of old-growth forests estimated at more than 108,000 ha. It is worth recalling the dramatic increase in secondary forests. The overall assessment of forest dynamics provides information on increasing deforestation over the two periods. Indeed, 14,983 ha of forests were deforested between 2003–2010 and over 37,000 ha between 2010–2016. As a result, the annual rates of deforestation almost tripled between 2003–2016. They went from 0.05% to 0.14% between 2003 and 2016.

**Table 5.** Areas and annual rates of deforestation between 2003–2010 and 2010–2016.

| Forest Type | Forest Areas | | | | | | Deforested Areas | | | |
|---|---|---|---|---|---|---|---|---|---|---|
| | 2003 | | 2010 | | 2016 | | 2003–2010 | | 2010–2016 | |
| | Ha | % | Ha | % | Ha | % | DA | Td | DA | Td |
| Pf | 3,801,767 | 91.75 | 3,751,719 | 91.73 | 3,643,399 | 89.28 | 50,048 | 0.19 | 108,319 | 0.42 |
| Sf | 178,472 | 5.83 | 213,538 | 5.28 | 284,351 | 6.91 | −35,065 | −2.56 | −70,813 | −4.09 |
| Total | 3,980,240 | 97.58 | 3,965,257 | 91.73 | 3,927,751 | 96.19 | 14,983 | 0.05 | 37,505 | 0.14 |

Td = Annual rate of deforestation in percentage; DA = Deforested area in hectares.

### 3.2.1. Historical Transitions

Table 6 summarizes the transitions observed between 2003 and 2016. From a global point of view, the historical dynamics of the landscape occur to the detriment of old-growth forest s over the entire observation period. Old-growth forests decrease by 2.69% compared to the proportion of 2003. They occupy from 91.75% in 2003. They barely represent 89% in 2016. In addition, secondary forests are experiencing an increase in area. They increased from 5.83% in 2003 to 6.91% of the total landscape area in 2016. This increase in secondary forests results from the conversion of old-growth forest s into secondary forests (2.18%) and non-forests into secondary forests (1.14%). Furthermore, the non-forest class increased by 26%, from 2.12% in 2003 to 3.50% of the total area of the landscape in 2016. The proportion of the landscape occupied by forests in 2003 and converted to non- forest in 2016 are estimated at 2.52% of the total area of the landscape. Indeed, secondary forests are the most affected by the changes. In terms of stability, the old-growth forest class shows great stability. In addition, non-forests are very fluctuating. Indeed, they show a stability of 46% compared to their proportion of 2003. The comparison of the proportions of land use in

2003 with those of 2016 does not reveal any significant changes in the composition of land use occupation. (X-squared = 0.46; df = 3; *p* = 0.93).

**Table 6.** Matrix of transitions between 2003 and 2016.

| 2003–2016 | | Land Use in 2016 | | | | Total 2003 |
|---|---|---|---|---|---|---|
| | | **Pf** | **Sf** | **Nf** | **Ww** | |
| Land use in 2003 | Pf | 87.66 | 2.18 | 1.90 | 0.00 | 91.75 |
| | Sf | 1.61 | 3.59 | 0.62 | 0.00 | 5.83 |
| | Nf | 0.00 | 1.14 | 0.97 | 0.00 | 2.12 |
| | Ww | 0.00 | 0.00 | 0.00 | 0.31 | 0.31 |
| | Total 2016 | 89.28 | 6.91 | 3.50 | 0.31 | 100 |

### 3.2.2. Deforestation Effort between 2003 and 2016

The smallest deforestation spot is 0.06 ha for all periods and for both classes of forest cover. On the other hand, the largest event of deforestation was estimated at 1007 ha over the period 2010–2016 in old-growth forest s. In addition, between 2003–2010, the biggest spot of deforestation in old-growth forest covers an area of 756 ha. The average area of deforestation plots in old-growth forest is estimated at 1 ha and 1.6 ha, respectively between 2003–2010 and between 2010–2016. Indeed, the area of deforestation spots observed in old-growth forest does not change significantly. depending on the observation period (*p*-value = 0.39). However, in secondary forest, the average area of deforestation is estimated at 1 ha between 2003–2010 and 0.7 ha between 2010–2016. The largest deforestation spot is estimated at 240 ha between 2003–2010 and at 410 ha between 2010–2016. There is also no significant difference between the areas of deforestation in secondary forest between the two periods (*p*-value = 0.54). For the entire observation period (2003–2016), the average area of deforestation in old-growth forest is 1.3 and 0.8 ha in secondary forest. Comparison of the deforestation spots in old-growth forest with those observed in secondary forests reveals a significant difference between the areas of deforestation spots observed in these two forest types (*p*-value = 0.04). Old-growth forest appears to be more vulnerable to deforestation than secondary forest. Taken as a whole, the deforestation spots observed between 2003–2010 seem to be smaller than those observed between 2010 –2016. Their average area is 1 ha between 2003–2010 and 1.2 ha between 2010–2016. Indeed, there is no significant difference between the areas of the deforestation spots over the two periods (*p*-value = 0.08). Figure 4 shows the variation in the areas of deforestation spots according to the type of forest cover. The diamond represents the mean.

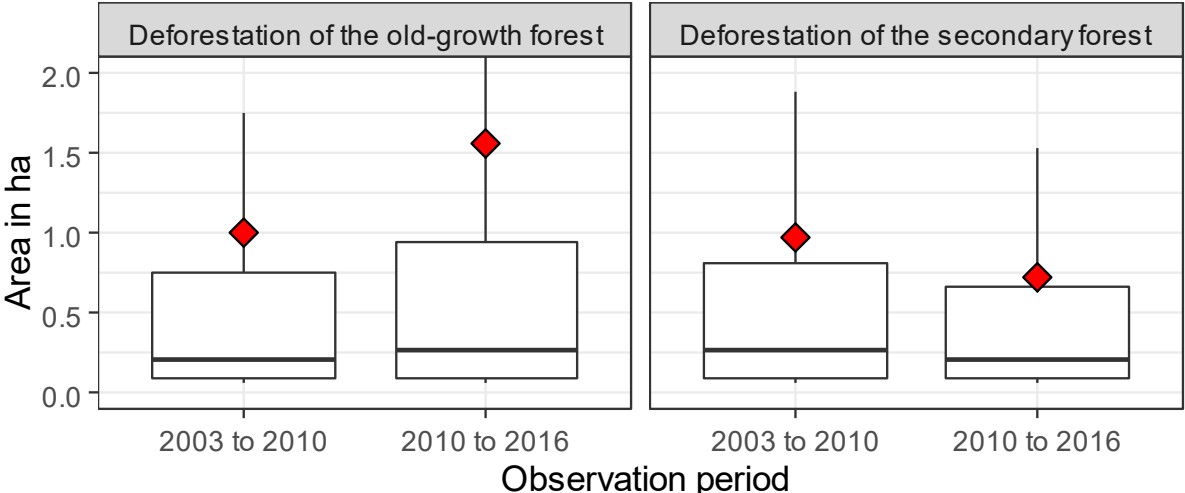

**Figure 4.** Variation in deforestation areas between 2003 and 2016.

Figure 5 illustrates these changes observed between 2003–2016. Visual analysis reveals that the landscape seems to be more affected by deforestation in the Southeast. Deforestation forms a disturbance gradient linked to the road and to major centers. The area of the wildlife reserve seems to be more stable. Indeed, this variability in forest deforestation seems to be a function of land use (macro-zones).

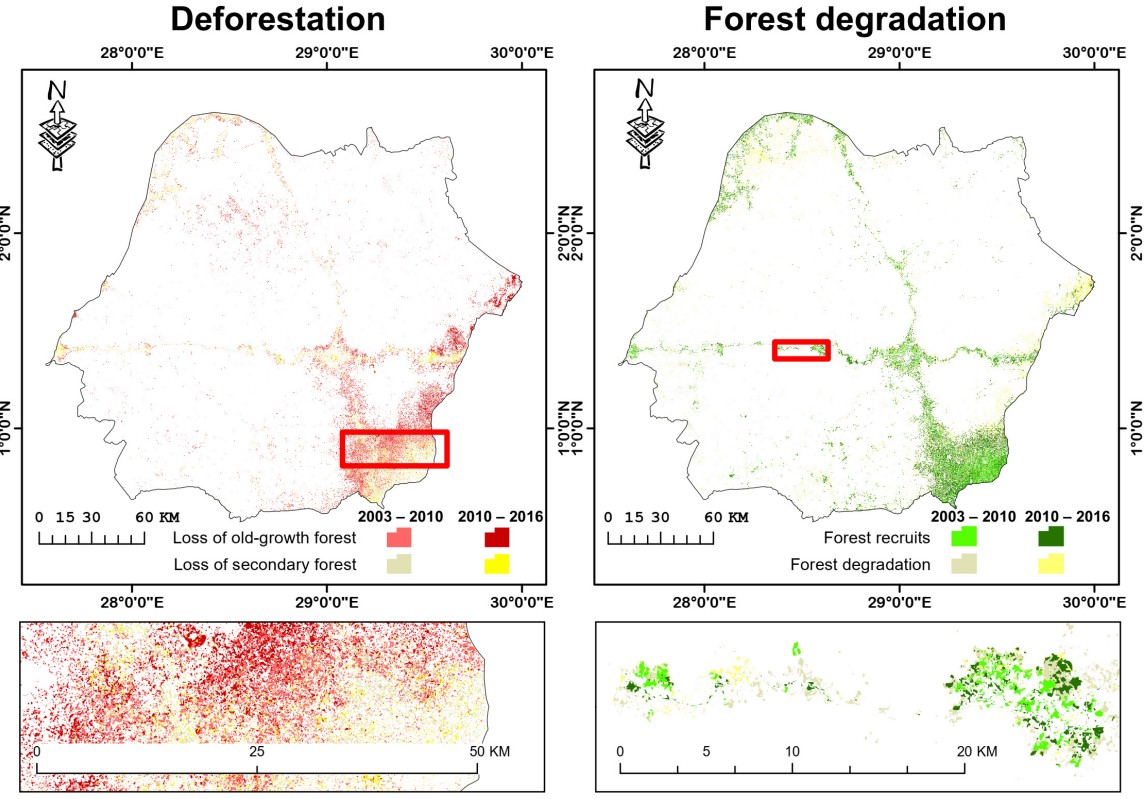

**Figure 5.** Mapping of forest cover changes: deforestation (**left**) and forest degradation (**right**).

### 3.3. Future Trajectories of Deforestation

3.3.1. Validation of the Model in 2014

The comparison of the changes observed and predicted between 2003 and 2014 made it possible to validate the simulation model of deforestation. The results of error budgeting and correct prediction reveal that 88.1% of the pixels in the landscape are correct due to observed and predicted consistency (Correct rejections [N]). Additionally, the correct pixels due to an observed and predicted change (Hits [H]) represent 4.25% of the pixels in the landscape. On the other hand, the errors due to a constancy observed but predicted to be changed (False alarms [F]) amount to 1.67% of the pixels in the landscape. The errors due to a change observed but predicted as constant (Misses [M]) are 5.42% (Figure 6). The total observed changes (OC = M + H) are 9.66% while the total predicted changes (PC = F + H) were underestimated with 5.92%.

The accuracy of the global prediction of changes across the entire landscape indicates that the quantity errors (Q = |F − M|) are estimated at 3.75% of the landscape pixels while the allocation errors [A = (F + M) − Q] represent only 3.34% pixels of the landscape. Therefore, the total error (Q + A) is 7.09% pixels of the landscape.

Figure 7 gives a spatial overview of the distribution of errors and correct predictions.

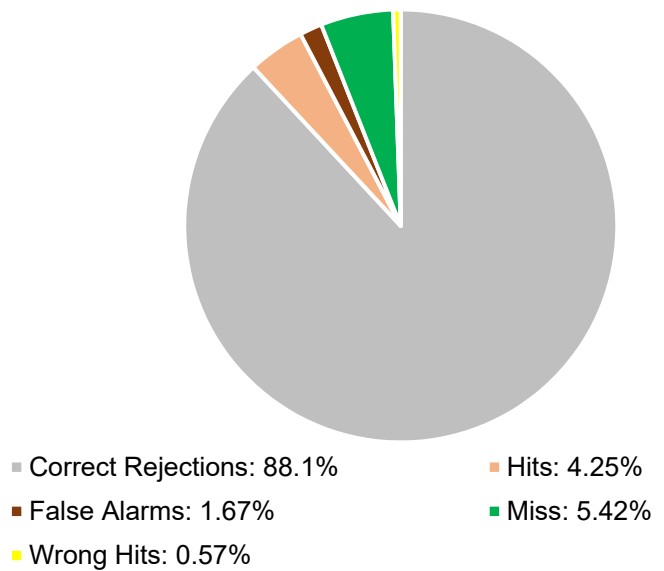

**Figure 6.** Budgeting for errors and correct predictions.

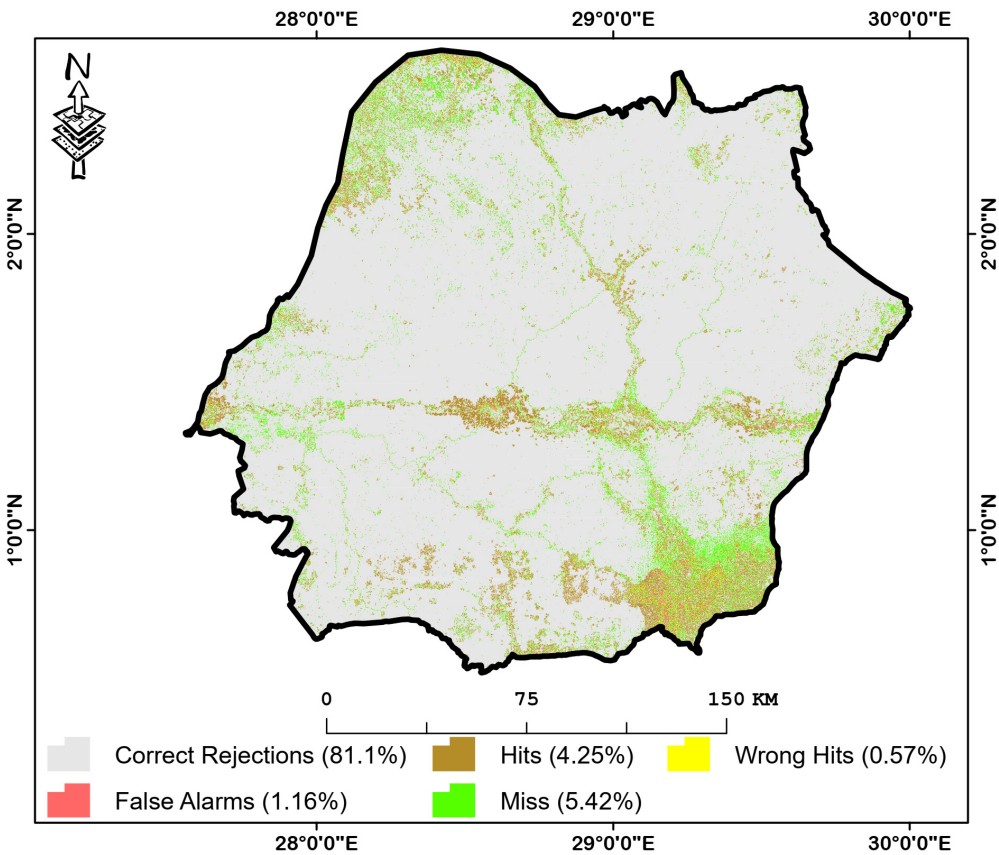

**Figure 7.** Map of errors and correct predictions.

The landscape observed in 2014 consists of 90.20% of old-growth forests, 6.16% of secondary forests, and 3.36% of non-forests and 0.28% of water. The concordance of simulated and observed land cover is estimated at 84.87% of the landscape for old-growth forests, 3.54% of the landscape for secondary forests and 1.37% of the landscape for non-forests. In addition, the landscape simulated in 2014 consists of 87.42% of old-growth forests, 8.17% of secondary forests, 4.11% of non-forests and 0.28% of water. The model

seems to underestimate old-growth forests. It also overestimates secondary forests, non-forests and water (Table 7).

**Table 7.** Comparison between observed and simulated land use.

| Observed–Simulated | | Simulated Land Use in 2014 | | | | Total Observed |
|---|---|---|---|---|---|---|
| | | **Pf** | **Sf** | **Nf** | **Ww** | |
| | Pf | 84.87 | 3.64 | 1.68 | 0.00 | 90.20 |
| Observed | Sf | 1.55 | 3.54 | 1.06 | 0.00 | 6.16 |
| land use in | Nf | 0.99 | 0.99 | 1.37 | 0.00 | 3.36 |
| 2014 | Ww | 0.00 | 0.00 | 0.00 | 0.29 | 0.28 |
| | Total simulated | 87.42 | 8.17 | 4.11 | 0.29 | 100 |

### 3.3.2. Future Trajectories of Deforestation

The combination of the transition matrix adapted to the different BAU, SME and REG scenarios with the transition potential maps and explanatory factors has made it possible to establish regular prospective monitoring until 2061 and the evolving statistics of land use areas (Figures 8–10). In the BUA scenario, the dynamic future of the landscape will come at the expense of old-growth forests (Figure 8).

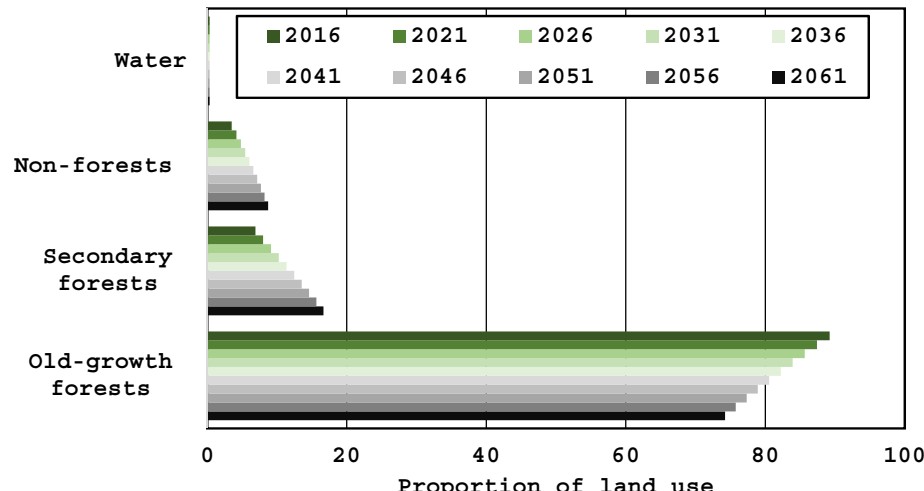

**Figure 8.** Future evolution of the composition of the occupation according to the trend scenario.

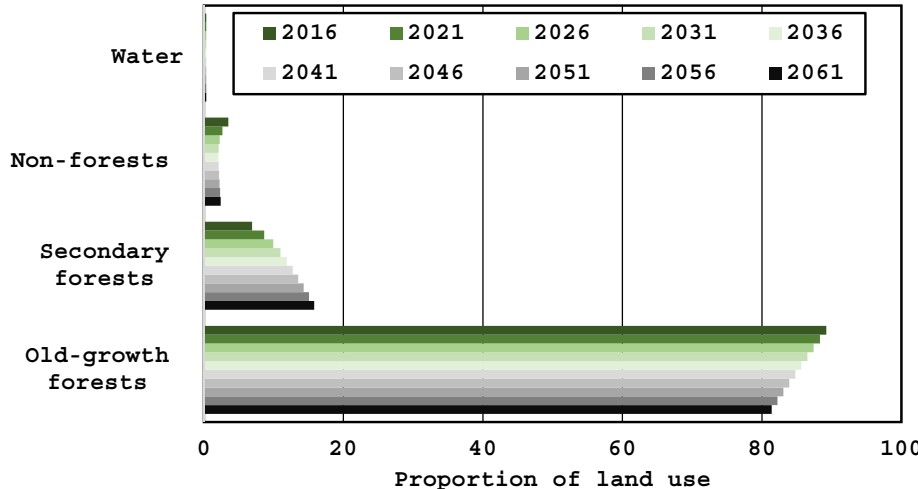

**Figure 9.** Future evolution of the composition of the occupation according to the scenario of SEM.

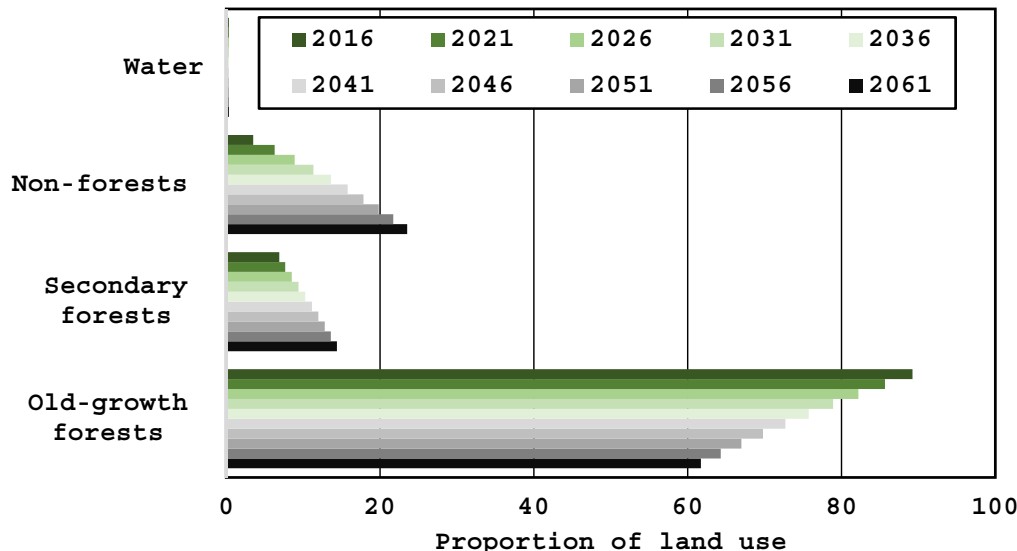

**Figure 10.** Future evolution of the composition of the occupation according to the REG scenario.

In the SEM scenario, the dynamics of land use will benefit forests (Figure 9). Considering the changes to be observed between 2016 and 2061, 8.84% of the proportion of the landscape occupied by old-growth forests will come from secondary forests (7.86%) and non-forests to secondary forests (2.73%). Furthermore, the non-forest class will experience a decrease in future years. They represent 3.50% of the proportion of the landscape in 2016. They will cover only 2.71% of the landscape by 2061. The proportion of the landscape that will remain unchanged is estimated at 87.03% of the total area of the landscape (respectively, 80.11% covered by old-growth forests, 5.95% by secondary forests, 0.66% non-forest and 0.31% water). Table 8 shows the transitions obtained in the SEM scenario between 2016 and 2061.

**Table 8.** Transition matrix in the SEM scenario between 2016 and 2061.

| 2016–2061 | | Land Use in 2061 | | | | Total 2016 |
|---|---|---|---|---|---|---|
| | | **Pf** | **Sf** | **Nf** | **Ww** | |
| Land use in 2016 | Pf | 80.11 | 7.86 | 1.30 | 0.00 | 89.28 |
| | Sf | 0.21 | 5.95 | 0.75 | 0.00 | 6.91 |
| | Nf | 0.12 | 2.73 | 0.66 | 0.00 | 3.50 |
| | Ww | 0.00 | 0.00 | 0.00 | 0.31 | 0.31 |
| | Total 2061 | 80.44 | 16.54 | 2.71 | 0.31 | 100 |

The change in the composition of land cover in the SEM scenario between 2016 and 2061 is illustrated in Figure 9.

In the rapid economic growth scenario, the future dynamics of the landscape will come at the expense of forests (Figure 10).

Between 2016 and 2061, 28.84% of the proportion of the landscape occupied by old-growth forests will be converted to secondary forests (12.62%) and non-forests (17.52%). The non-forest class will experience a dramatic increase in the years to come. It represents 3.51% of the proportion of the landscape in 2016. It will cover more than 25.25% of the landscape by 2061. The proportion of the landscape that will remain unchanged is estimated at 64.22% of the total area landscape (respectively, 59.13% covered by old-growth forests, 1.92% by secondary forests, 2.86% non-forest and 0.31% water). Table 9 shows the transitions obtained in the REG scenario between 2016 and 2061.

**Table 9.** Transition matrix in the REC scenario between 2016 and 2061.

| 2016–2061 | | Land Use in 2061 | | | | Total 2016 |
|---|---|---|---|---|---|---|
| | | **Pf** | **Sf** | **Nf** | **Ww** | |
| Land use in 2016 | Pf | 59.13 | 12.62 | 17.52 | 0.00 | 89.28 |
| | Sf | 0.11 | 1.92 | 4.87 | 0.00 | 6.91 |
| | Nf | 0.07 | 0.58 | 2.86 | 0.00 | 3.50 |
| | Ww | 0.00 | 0.00 | 0.00 | 0.31 | 0.31 |
| | Total 2061 | 59.31 | 15.13 | 25.25 | 0.31 | 100 |

The comparison of the proportions of land use simulated in 2061 with those observed in 2016 shows significant differences in two different scenarios: BAU (X-squared = 16.46; df = 3; $p$ = 0.03) and REG (X-squared = 17.25; df = 3; $p$ = 0.001). Moreover, the simulated occupation of the SEM is not statistically different from that observed in 2016 (X-squared = 20.71; df = 3; $p$ = 0.06).

## 4. Discussion

### 4.1. Historical and Future Trajectories of Deforestation

The dynamics of land use in the study area are characterized by deforestation and forest degradation. Deforestation observed in the Ituri-Epulu-Aru landscape shows significant differences between periods, forest types and macro-zones (protected area, sustainable management zone for natural resources and extraction zone). Indeed, before 2010, the annual rate of deforestation was relatively low (0.05%) and the average area of deforestation spots was 1 ha. It more than doubled between 2010–2016 reaching 0.14% per year and the average area of deforestation spots increased by 1.2 ha. The significant decrease over time in forest area confirms the hypothesis of continual anthropization of Ituri's forests. However, comparing deforestation rates by period does not reveal any significant difference. Likewise, in all cases, the average area of deforestation spots is not significantly different over the two periods, which shows that there is no "period" effect on deforestation rates.

Considering land use, the differences in annual deforestation rates are very large ranging from 0.02 to 3.05% for the period 2003–2010 and from 0.1 to 3.20% for the second observation period. At the landscape level, these rates remain below the national average of 0.22% per year [1]. Moreover, except in the OWR, these rates are above this average, particularly in the second period. Several authors share the same opinion that deforestation is increasing in the majority of forests [2,29].

Comparison of key deforestation figures obtained in this study with those of other similar studies should be done with caution since the methodologies and data used are not always compatible. FACET [2] estimates the area of old-growth forests in 2010 at 3,843,218.88 ha. This area is slightly less than that obtained in the present study. Some scenes used may be different. Statistics from FACET [2] reveal increasing rates of deforestation, a trend shared by our results. Furthermore, Lusana et al. [14] estimate this area at 4,049,204 ha in 2003 and 3,997,690 ha in 2010, i.e., a loss of 51,514 ha between 2003 and 2010. Note that this latest study is based on the mapping materials of Hansen et al. [13] who overestimate the forest area [25]. The main reason given by these researchers is that the scale of analysis does not allow a good definition of the forest. Thus, it is possible that certain wastelands are confused with forests. This explains why the estimates of Lusana et al. [14] seem to exceed those carried out in this study.

Deforestation rates observed in the Ituri-Epulu-Aru landscape remain relatively low compared to other regions of the country, such as in the Bombo-Lumene reserve located not far from Kinshasa (0.46% per year between 2000 and 2015), the Yangambi Biosphere Reserve (4.5% between 2003 and 2016) [29] and very low compared to tropical America (0.51%) or Tropical Asia (0.58%) [58].

*4.2. Simulation of Deforestation*

This article provides useful information for those who wish to discuss a model that can be replicated for other territories affected by deforestation and changes in natural and anthropogenic forest structure. Fieldwork identified agriculture, forestry, infrastructure, demographic factors, socio-political factors, economic factors and biophysical factors. Among the variables retained, the distance from rural complexes, distance from national roads, artisanal mining and distance from major centers seems to play an important role in view of the main changes observed between 2003 and 2016. These results are similar to those of several authors [11,59].

The development of images from trendy and contrasting prospective scenarios will promote the identification of areas with socio-environmental issues concerning on the one hand the living environment of Pygmy communities and on the other hand the preservation of old-growth forests. For decades, primary and secondary forests have given way to crops in agricultural areas [59]. The use of these deforested areas makes it possible to benefit from new fertile land and therefore to increase agricultural production.

In summary, the trend of regression of the forest landscape in favor of culture and urban spaces has been observed for several decades [2]. This is then done at the expense of urban and village centers but also along the main communication axes (road network, network of tracks) [3,11]. Moreover, this degradation mainly impacts old-growth forests [28]. Suddenly, deforestation leads to a loss of biodiversity due to the destruction of many natural habitats [60]. The different prospective scenarios designed here take into account the different socio-economic activities developed in the study environment.

In general, forest dynamics are regressive although secondary forests are increasing. The trend scenario (BAU) suggests an alarming deforestation in the next four decades, which makes it possible to verify the fourth hypothesis. In this BAU scenario, both non-forests and secondary forests have increased. Indeed, this increase could be explained by the increase in population and therefore the need for food and housing. In compensation for this strong demand for land, there is a reduction in the area of old-growth forests. These results are corroborated by those of Samie et al. [61] obtained in Punjab (Pakistan).

The catastrophic scenario (REG) predicts that natural plant formations will regress in favor of anthropogenic ones. The sustainable environmental management (SEM), which combines both the preservation of plant cover with agricultural activities, empowers the state in its role of controlling deforestation and subsidizing domestic gas to replace fuelwood. This is similar to the densification scenario developed by Lajoie and Hagen-Zanker [62] which encourages the preservation of forests and limits urban sprawl in Reunion Island by 2031.

The validation of the model constitutes a first step in the prospective modeling of forests by 2061. The prospective model designed presents conclusive results and seems to be able to better take into account the evolution trends, the latter, by its unsupervised character.

The prediction model developed in this study to estimate the quantities of land cover changes produced values close to reality. In fact, it confirms, on the whole, the trends in land use. Nevertheless, it presented difficulties in predicting the changes that took place between 2003 and 2014. This is linked to the high observed and simulated constancy which was 88% at the landscape scale. This means that our analysis, both at the landscape level and at the level of land cover classes, highlights interesting results but which should be qualified. Moreover, there are fewer false alarms than failures, indicating that the simulated change is less than the baseline change. The quantity component does not indicate whether the false alarms are less than the misses or vice versa. The quantity component is about the same size as the allocation component. If our false alarms are less than your errors, it may be because the rate of change during the calibration time interval is slower than the rate of change during the validation time interval.

Overall, the observation of errors reveals that they are localized near the non-forests observed. The limitation of the model lies mainly in the fact that there are other variables that may explain the changes in land cover and use. These are, for example, political

and institutional factors such as poverty, unemployment, conflicts and the forest code, demographic factors such as migration and population distribution, cultural factors (household consumption) and economic factors (cost of labor and capital). The addition of these additional variables was limited by their non-quantifiable nature and their unavailability in digital format [19,27,55,63]. In addition to the variables used.

## 5. Conclusions

This article aimed to analyze deforestation in the Ituri-Epulu-Aru landscape. This article provides valuable information on deforestation and forest degradation patterns. The results obtained confirm the trend towards deforestation. Although the landscape has seen a slight increase in the area of secondary forests, that of old-growth forests has declined significantly. Taken as a whole, forests are shrinking as a result of the unsustainable land use pattern characterized by shifting slash-and-burn agriculture with increasingly shorter fallows. The great concern lies in maintaining priority habitats for the biodiversity of this landscape. Our model predicts an increase in secondary forests over the entire landscape studied. This increase is not good news; indeed, it is indicative of a strong deterioration due, in particular, to subsistence activities.

Taking into account the results obtained, we propose that the landscape management consortium initiates local environmental and social management plans around hot zones of deforestation, in particular around Mambasa and Walese-Vonkutu. Land use should favor the restoration of severely degraded landscapes while highlighting sustainable development approaches (agro-ecology, renewable energies, etc.) and biodiversity conservation (particularly in sensitive areas). Raising awareness and improving the agrarian system through agroforestry techniques (particularly agro-forests, which are particularly suited to the region) must be at the center of strategies for the creation and development of village secondary forests, a pledge of the sustainable management of natural resources from which the populations will be able to obtain forest products for their usual needs.

It is also recommended that future analyzes assess the influence of this deforestation on the climate by quantifying the associated emissions. It would be interesting to say the impact of this loss of forests on the well-being of neighboring populations in order to further inform policy choices.

**Author Contributions:** Conceptualization, J.M.K., O.M.K., R.N. and L.M.; methodology, J.M.K., O.M.K., T.M.B. and L.M.; software, J.M.K.; validation, J.M.K., O.M.K., T.M.B., R.N. and L.M.; formal analysis, J.M.K.; investigation, J.M.K. and A.B.L.; resources, J.M.K., M.S., N.M., T.M.B. and L.M.; data curation, J.M.K.; writing—original draft preparation, J.M.K.; writing—review and editing, J.M.K., O.M.K., M.S., N.M., T.M.B., R.N.; visualization, J.M.K.; supervision, O.M.K., R.N., J.-P.M.M. and L.M.; project administration, O.M.K. and L.M. All authors have read and agreed to the published version of the manuscript.

**Funding:** This research received no external funding.

**Institutional Review Board Statement:** Not applicable.

**Informed Consent Statement:** Not applicable.

**Data Availability Statement:** Data presented in this study are available on request from the corresponding author. The data are not publicly available because they are part of ongoing research.

**Acknowledgments:** We warmly thank Victor Kadiata for having the Satellite Observatory of the Forests of Central Africa (OSFAC) for ensuring the download of satellite images. We sincerely thank Alastair McNeilage and Toussaint Molenge (CARPE) for recommending us to SOFCA. Big thanks to Salomon Dyoma Ezadri, Alimengo Paul, Atimo Michel, Kimate Henry, Landasi Tsavurume, M'vuge Bijoux, Mamimolo Jean-Marie, Mumbere Jackson, Ngenda Elvis, Mundele Oscar, Yambwa Gloire for assistance in the field. We also thank the former Chief Curator/Site Manager of the Okapis Wildlife Reserve and Directors of Parks and Reserves in DR Congo Paulin TSHIKAYA and the Curator/Head of Research and Biomonitoring Program at the Okapis Wildlife Reserve Balimbaki Aimé for access to the study site. We thank Rosemarie Ruf for its support. We are very grateful to Robert Gilmore

Pontius and Thomas Bilintoh for their valuable comments on the methodology and particularly on the validation of the simulation model.

**Conflicts of Interest:** The authors declare no conflict of interest.

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
