# Peer review of "Historical Changes and Future Trajectories of Deforestation in the Ituri-Epulu-Aru Landscape (Democratic Republic of the Congo)"

_land, doi:10.3390/land10101042_

Round 1
Reviewer 1 Report
The manuscript is fine, but lots of ideas are merging. Therefore, I would like to suggest “please write database, results in a systematic way.” And please articulate the novelty and the significance of the paper in the abstract, discussion, and conclusion sections. Also, please avoid unnecessary statements or general comments in the manuscript.
Please make sure all dash, font size, space, a hyphen, en dash, and capital words would be appropriate throughout the manuscript.
Please make sure the font size in figures and tables.
Please follow the journal guideline. Please check references.
Abstract:
What is DINAMICA EGO platform?
1. Introduction
Line 52-53: Please correct sentence (.?)
Line 84-85: Please cited the figure according to journal guidelines.
Line 59-61 AND Line 91-93: No need to repeat the sentences.
Line 118: Need to add detail information (data/ path row/year/ etc.) of Landsat TM, ETM + and OLI images.
Line 152: Table 1. Description of land use classes. Need to write the sources of this table or justify how Author(s) define classification scheme of different landcover classes?
Line 158: Figure 2 is great, but section b in figure 2 is still confusing. Could you please revise it (systematic way) for better understanding?
Line 164: R package “Random Forest” Need to describe more (figure?) about Random Forest workflow in R package for better understanding.
2.3.2.1. Selection of variables
Need to add method description for selecting the variables and their independence for the deforestation modeling. Did Author (s) apply multicollinearity analysis between factors before applying the model? https://iopscience.iop.org/article/10.1088/1742-6596/949/1/012009/pdf
Line 201: Table 2 Mining square (Mining) and Distance to mining squares (d_mining) – what is the difference?
Don’t you think you should add all figures and data sources of these explanatory variables of deforestation in the manuscript for better understanding?
2.3.2.2. Transitions
Need to revise this section, not well presented.
Line 208-211: In this study, the transitions of interest were as follows: a) old-growth forest to secondary forest, b) old-growth forest to non-forest, c) secondary forest to old-growth forest, and d) secondary forest to non-forest. Did the author (s) use transitions forest as evidence to apply the WoE method to prepare a probability map of deforestation?
2.3.2.3. Exploratory analysis of the data
Line 213: weights of proof (WoE) or Weights of Evidence?
Lots of information in this section but not well presented.
For example, Line 224-225: no description of these methods?
And Line 230-231: No information and results about 20 explanatory variables of deforestation. Which variables were highly correlated (> 0.5 value?)
3.2. Analysis of historical changes of deforestation between 2003 and 2016
How about before 2003 and the current situation (2020) of land use in the study area? Is it possible to add some facts or figures for better understanding?
Table 4 annual rates of deforestation between 2003 - 2010 and 2010 – 2016: Recommended article 10.3390/environments4020034
Author Response
Abstract: What is DINAMICA EGO platform? DINAMICA EGO is the modeling platform used in this study. 1. Introduction Line 52-53: Please correct sentence (.?) The sentence has been corrected. The period that made the sentence incoherent has been removed. Line 84-85: Please cited the figure according to journal guidelines. The figure was cited according to the guidelines of the journal. Line 59-61 AND Line 91-93: No need to repeat the sentences. Lines 91-93 have been deleted to avoid redundancy. Line 118: Need to add detail information (data/ path row/year/ etc.) of Landsat TM, ETM + and OLI images. This information is not available for the annual composites because they are a multitude of synthetic images processed in Google Earth Engine. These images are averages calculated between the first and last day of the year over the entire Congo Basin. They are organized in tiles of one degree. It should be noted that the study area is located on the scenes (path row) 174/058, 174/059, 174/060, 173/058, 173/059, 173/060. Line 152: Table 1. Description of land use classes. Need to write the sources of this table or justify how Author(s) define classification scheme of different landcover classes? We have taken care to add the sources Line 158: Figure 2 is great, but section b in figure 2 is still confusing. Could you please revise it (systematic way) for better understanding? This figure has been reworked to make it more understandable. First, section a): The study landscape is over 80% dominated by primary forest for all dates. Therefore, it is unlikely to result in an over all of less than 80%. This is why we have reworked section a of figure 2. Also, on section b of the same was simplified. Line 164: R package “Random Forest” Need to describe more (figure?) about Random Forest workflow in R package for better understanding. We have added key references that explain the random forest package in R. 2.3.2.1. Selection of variables Need to add method description for selecting the variables and their independence for the deforestation modeling. Did Author (s) apply multicollinearity analysis between factors before applying the model? https://iopscience.iop.org/article/10.1088/1742-6596/949/1/012009/pdf Nous avons effectivement appliqué une analyse de multicollinéarité entre les facteurs avant d'appliquer le modèle. Il est nécessaire d'ajouter une description de la méthode de sélection des variables et de leur indépendance pour la modélisation de la déforestation. L'auteur (les auteurs) a-t-il (ont-ils) appliqué une analyse de multicollinéarité entre les facteurs avant d'appliquer le modèle ? https://iopscience.iop.org/article/10.1088/1742-6596/949/1/012009/pdf Line 201: Table 2 Mining square (Mining) and Distance to mining squares (d_mining) – what is the difference? The mining square is the basic unit of the mining or quarrying perimeter as defined by the cadastral grid of the National Territory. The distance to the mining square is the smallest distance between the mining square and the study area. Don’t you think you should add all figures and data sources of these explanatory variables of deforestation in the manuscript for better understanding? All figures and data sources for these deforestation explanatory variables have been added in the manuscript for better understanding. 2.3.2.2. Transitions Need to revise this section, not well presented. Line 208-211: In this study, the transitions of interest were as follows: a) old-growth forest to secondary forest, b) old-growth forest to non-forest, c) secondary forest to old-growth forest, and d) secondary forest to non-forest. Did the author (s) use transitions forest as evidence to apply the WoE method to prepare a probability map of deforestation? This section has been revised: Transitions modeled in this study include: a) transition from old-growth to secondary forest (degradation), b) transition from old-growth to non-forest (deforestation), c) transition from secondary forest to old-growth (maturation), and d) transition from secondary forest to non-forest (deforestation) 2.3.2.3. Exploratory analysis of the data Line 213: weights of proof (WoE) or Weights of Evidence? weight of evidence. This confusion has been removed in the manuscript 3.2. Analysis of historical changes of deforestation between 2003 and 2016 How about before 2003 and the current situation (2020) of land use in the study area? Is it possible to add some facts or figures for better understanding? Unfortunately, we cannot provide information before 2003 and the situation in 2020 because of the lack of ground truths in 2020 because satellite images exist. Table 4 annual rates of deforestation between 2003 - 2010 and 2010 – 2016: Recommended article 10.3390/environments4020034 We reworked Table 4 annual rates of deforestation between 2003 - 2010 and 2010 - 2016 as recommended by the reviewer (10.3390/environments4020034). But we did not use the same equations because the objectives differ. The reviewer's recommendation (10.3390/environments4020034) Our review |

Reviewer 2 Report
The article Historical changes and future trajectories of deforestation in 2the Ituri-Epulu-Aru landscape (Democratic Republic of Congo) is focused on the realization of three contrasting scenarios: starting from the hypothesis of the absence of new economic or environmental policies, a scenario in which legislation and government subsidies encourage the emergence of forestry and a “socio-economic” scenario with tendency towards disaster.
This article has a high value for the scientific field in which it falls and brings useful information for those interested in discussing a model that can be replicated for other territories affected by deforestation and changes in the natural and anthropogenic structure of the forest.
The article may be published after minor changes recommended in the following:
In line 113 it is not necessary to specify what figure 1 refers to (Figure 1 presents the geographical and topographicalcontext of the study area) this is understood from the name of figure 1.
Similar for row 357 (Figure 7 gives a spatial overview of the distribution of errors and correct predictions) and 386-387. For figure 1 I recommend the use of a color ramp that respects the cartographic principles. This ramp of colors from green to red is generally used for land slope or in risk studies, etc. For the altitude of the relief in the range 425-4290 m it would be useful to use colors from green to intense brown.
In the Methodology section you state that the data taken for 2016 corresponds to the last year for which the data are available. No more recent data available?
Conflicts of Interest: The authors declare no conflict of interest are in the rowd 522 and 543 Please delete one of them.
Best regards,
Author Response
In line 113 it is not necessary to specify what figure 1 refers to (Figure 1 presents the geographical and topographical context of the study area) this is understood from the name of figure 1. This sentence has been deleted for more coherence. Similar for row 357 (Figure 7 gives a spatial overview of the distribution of errors and correct predictions) and 386-387. For figure 1 I recommend the use of a color ramp that respects the cartographic principles. This ramp of colors from green to red is generally used for land slope or in risk studies, etc. For the altitude of the relief in the range 425-4290 m it would be useful to use colors from green to intense brown. Figure 1 has been reworked following cartographic principles using a color ramp that corresponds to the altitude. In the Methodology section you state that the data taken for 2016 corresponds to the last year for which the data are available. No more recent data available? Data are available for years after 2016. Indeed, the year 2016 was chosen in line with a field data collection campaign. Lines 131-134 have been corrected In addition, 2016 was chosen in alignment with a field data collection campaign. And 2010 is the year that roughly halves the observation period (2003 and 2016). The year 2014 was chosen for the validation of the spatialized prospective model. Indeed, 2014 is relatively close to 2010 and 2016 and far enough away from 2003; an ideal time step for validation [27-29]. Conflicts of Interest: The authors declare no conflict of interest are in the rowd 522 and 543 Please delete one of them. Line 543 has been deleted. |

Round 2
Reviewer 1 Report
I think the manuscript can be accepted.
This manuscript is a resubmission of an earlier submission. The following is a list of the peer review reports and author responses from that submission.